# Factors Associated with Reduction in Physical Activity during the COVID-19 Pandemic in São Paulo, Brazil: An Internet-Based Survey Conducted in June 2020

**DOI:** 10.3390/ijerph182111397

**Published:** 2021-10-29

**Authors:** Gabriela Togni, Paulo José Puccinelli, Taline Costa, Aldo Seffrin, Claudio Andre Barbosa de Lira, Rodrigo Luiz Vancini, Douglas De Assis Teles Santos, Pantelis Theodoros Nikolaidis, Beat Knechtle, Marilia Santos Andrade

**Affiliations:** 1Department of Physiology, Federal University of São Paulo, São Paulo 04021-001, Brazil; gabizinha_togni@hotmail.com (G.T.); paulopuccinelli@hotmail.com (P.J.P.); costa.taline@gmail.com (T.C.); netoseffrin@gmail.com (A.S.); marilia1707@gmail.com (M.S.A.); 2Human and Exercise Physiology Division, Faculty of Physical Education and Dance, Federal University of Goiás, Goiânia 74690-900, Brazil; 3Center for Physical Education and Sports, Federal University of Espírito Santo, Vitória 29075-910, Brazil; rodrigoluizvancini@gmail.com; 4Faculty of Physical Education, State University of Bahia, Teixeira de Freitas 45992-255, Brazil; datsantos@uneb.br; 5School of Health and Caring Sciences, University of West Attica, 122 43 Athens, Greece; pademil@hotmail.com; 6Institute of Primary Care, University of Zurich, 8091 Zurich, Switzerland; beat.knechtle@hispeed.ch

**Keywords:** COVID-19, physical activity, sedentarism, coronavirus, exercise

## Abstract

Background: The COVID-19 pandemic negatively affected physical activity levels. This study investigated the factors associated with the change in physical activity level in Brazilians residing in the city of São Paulo. Methods: A self-administered questionnaire, addressing personal data, restriction level, education level, family income, daily working hours, and physical activity level, was answered by 2140 volunteers, of which 1179 were excluded because the answers were either incomplete or the respondents were not from São Paulo. The total number of participants selected was 961 (581 female and 380 male). Results: The physical activity level adopted prior to the pandemic period (*p* < 0.001) and family income (*p* = 0.001) correlated significantly with physical activity level reduction during the pandemic. The proportion of people who reduced their physical activity was greater among those who were very active than those who were active (adjusted prevalence ratio [aPR]: 0.65 [confidence interval (CI): 0.52–0.80]) or insufficiently active [aPR: 0.39 (0.18–0.82)]. The proportion of people who reduced their physical activity was greater among those who received a salary less than minimum wage (MW) than those who received a salary between three to six times minimum wage [(aPR: 0.50 (CI 0.35–0.70)] or more than 6 MW [(aPR: 0.56 (CI 0.40–0.79)]. Conclusions: A higher prevalence of Brazilians residing in the city of São Paulo reduced their physical activity who had a vigorous level of physical activity prior to the pandemic and who received less than a MW.

## 1. Introduction

COVID-19 is an infectious disease caused by severe acute respiratory syndrome coronavirus 2 (SARS-CoV-2) [1,2]. It was first detected in the city of Wuhan, China. There are several strategies to prevent coronavirus contagion, in addition to vaccination. The use of masks, maintaining social distancing [1] (which is facilitated by allowing the operation of only essential services in the areas of health, safety, and nutrition), have been considered as important measures.

Insufficient physical activity was observed as one of the social distancing isolation measures in most of the papers published on the subject [3,4,5]. Physical inactivity correlated with several adverse health outcomes, and approximately 3.2 million deaths each year were attributable to insufficient physical activity [6].

The physical inactivity scenario was extremely worrying in the COVID-19 pandemic period. Physical activity improves the immune system and the antiviral defense, reducing the risk, duration, and gravity of viral infection [1,7]. Physical activity also improves lung function and respiratory muscles [3]. This prepares the body for prolonged mechanical ventilation, which could be necessary for patients infected by the coronavirus [8,9]. Physical inactivity is one of the leading causes of several noncommunicable diseases such as diabetes, arterial hypertension, and obesity [1,10,11,12], which worsened the COVID-19 prognosis [13]. Higher body mass index (BMI) is a risk factor for contracting COVID-19 [14].

With this information in mind, physical activity programs have been consistently encouraged in the pandemic period, and home-based physical activity programs have received special prominence [15,16,17,18,19]. Numerous tips have been published for home-based physical activity and sedentary behavior interruption [20]. The impact of social distancing on physical activity level was not totally elucidated despite these recommendations being presented in scientific publications and via newspapers, the internet, TV, and radio.

The World Health Organization (WHO) considered some environmental factors as partially responsible for physical inactivity, such as violence, high-density traffic, low air quality, and the unavailability of parks, sidewalks, and sports/recreation facilities [6]. It was reasonable to assume that there were also several other potential factors that could influence the physical activity levels during the pandemic period. Despite the general recommendations for staying at home, the level of restriction adopted by each subject and each country was quite variable [21], and those who adopted stricter measures of social distancing may have had more difficulty in maintain their physical activity levels. The age of the subject might also affect social distancing. Considering that male older adults tended to have more risk factors for a worse prognosis of COVID-19 (older age and presence of noncommunicable diseases) [22], they may be more concerned with maintaining the level of physical activity than younger individuals.

Moreover, the level of physical activity adopted before the pandemic period could also be an intervening factor [23]. Although home-based exercise is being encouraged, people with a very intense level and/or high volume of physical activity may have difficulty in maintaining the intensity/volume in a restricted physical space.

Education, family income, and the number of daily working hours are all potential factors that influence the impact of social distancing on physical activity [24]. In a previous study, Pitanga et al. (2017) showed that people with a college education had a higher prevalence of leisure-time physical activity than those with an elementary or high school education. The same authors also showed that those with a monthly family income higher than 18 minimum wages also had a higher prevalence of leisure-time physical activity than those with a lower family income.

The knowledge of the factors associated with the reduction of physical activity during the pandemic period could help the Health Ministry and coaches to develop programs to encourage the practice of physical activity. These programs should target those who would be more vulnerable to the impacts of social distancing on the active lifestyle.

This study compared physical activity levels before and during the COVID-19 pandemic period of Brazilians residing in the city of São Paulo. The study also analyzed the level of association between the change of physical activity and restriction level, age, gender, prepandemic physical activity level, education level, family income, and daily working hours.

It was hypothesized that if the recommendation to maintain physical activity at home was respected, the physical activity level would decrease significantly in this pandemic period. This reduction in the level of physical activity would be greater among those adhering to more restricted measures of social distancing; those who presented a more vigorous physical activity level prior to the pandemic period; those with longer daily working hours; and those with lower education level and family income.

## 2. Materials and Methods

The first case of COVID-19 in Brazil was detected in late February 2020 [25] in São Paulo. Since then, the pandemic has expanded in the country, and by May 31, 2020, there were 514,992 confirmed cases and 29,341 deaths [26]. Because of the rapid increase in the number of new cases and the lack of certainty regarding the return to routine life and physical activity, the study was carried out during this period. This was a cross-sectional study conducted from 2 to 12 June, 2020. At this time, Brazil was adopting several measures to limit the COVID-19 spread (close all public services including gyms and parks and trade restricted to supermarkets, pharmacies, delivery restaurants, gas stations, and other critical services). Social distancing and other COVID-19–related measures, such as hand hygiene, use of masks, cleaning and disinfection of environments, and isolation of suspected and confirmed cases were adopted mostly at the state and city levels [25]. The participants were recruited through advertisements in social networks (WhatsApp, Instagram, and Facebook). A link to access the questionnaire, shared through the Google Forms digital platform, was sent.

### 2.1. Questionnaire

Participants answered a self-administered questionnaire divided into three sections. The first section included general data on email address (open-ended question), age (open-ended question), gender (multiple-choice question: male or female), body mass and height (open-ended questions), education level (multiple-choice question: incomplete or complete elementary level, incomplete or complete middle school level, incomplete or complete high school level, incomplete or complete college level, incomplete or complete graduation level), family income (multiple-choice question: less than 1 minimum wage, minimum wage between 1 and 2, minimum wage between 3 and 6, and more than 6 minimum wages), level of restriction adopted during the social distancing period (multiple-choice question: completely adhered to the social distancing measures, leaving home only for essential nonwork activities, leaving home also for work activities, did not adhere to social distancing isolation), and daily working hours (open-ended question). For the purpose of analysis, educational levels were grouped in four levels: middle level (complete middle school level and incomplete high school level), high school level (complete high school level and incomplete college level), college level (complete college level and incomplete graduate level), and graduate level (complete graduate level). Nobody answered incomplete or complete elementary school, incomplete or complete middle school. The minimum wage in Brazil corresponds to about 190 dollars (based on December 2020 values).

The second section comprised questions related to the current physical activity level (during the pandemic period). The International Physical Activity Questionnaire (IPAQ) was used for this purpose [27]. These questions address the regular practice of daily physical exercise and classify the individual according to the level of daily activity into very active (those who perform vigorous activities 5 days/week and ≥30 min per session or vigorous activities ≥3 days/week and ≥20 min per session + moderate activities ≥5 days/week and ≥30 min per session), active (those who perform vigorous activities ≥3 days/week and ≥20 min per session; or moderate activities ≥5 days/week and ≥30 min per session; or any combined activity: ≥5 days/week and ≥150 min/week (walking + moderate + vigorous), irregularly active A (those who perform physical activity, but it is insufficient to be classified as active because it does not comply with the recommendations regarding frequency or duration), irregularly active B (those who perform physical activity, but it is insufficient to be classified as irregularly active A, because it does not comply with either the frequency or duration recommendations), and not active (those who did not perform any physical activity for at least 10 continuous minutes during the week). For the purpose of analysis, scores from 1 to 5 were assigned to activity levels, where 1 referred to the lowest level of activity (not active) and 5 to the highest level of activity (very active).

The third section used the IPAQ to assess the level of physical activity of participants prior to the COVID-19 pandemic period. The questions and classifications used in this section were the same as in the previous section. For analysis purposes, a score of 0 was referred to as a reduction in the physical activity level, and a score of 1 was referred to as no reduction in the physical activity level (maintenance or increase).

### 2.2. Participants

Participants had to be over 18 years old and complete the entire questionnaire to be included in the study. All participants were informed about the aims of the research by reading a text prior to the application of the questionnaire. Informed consent was received from the participants.

A total of 2140 questionnaires were answered, of which 740 were excluded because the answers were incomplete, and 439 were excluded because they were from different states. Thus, the total number of participants selected was 961 (581 female and 380 male) residents from São Paulo. Characteristics of the participants are presented in Table 1. The sample from this study was part of a larger sample studied in Puccinelli et al. (2021).

### 2.3. Ethical Approval

The study followed the principles outlined in the Declaration of Helsinki and was approved by the Human Research Ethics Committee of the Federal University of São Paulo (UNIFESP, Brazil) (Approval number: 4.073.442).

### 2.4. Statistical Analysis

According to the Kolmogorov–Smirnov test, no variables presented a normal distribution. Categorical variables were expressed in absolute and/or percentage values. Numerical variables were expressed as median (interquartile interval). The Mann–Whitney test was used to verify differences between genders according to age, body mass, height, and BMI. The Wilcoxon signed rank test was used to verify differences between physical activity level prior and during the pandemic period. The McNemar test was used to identify at the level of physical activity which had a change in number between the two periods (dummy variables were created for each level of physical activity to enable dichotomous traits and make this comparison possible).

To identify the factors associated with decreased levels of physical activity, crude and multivariable analysis was performed with estimates of the prevalence ratios through Poisson regression. The crude models were constructed containing each of the independent variables and the response variable (decreased level of physical activity, yes or no). To calculate the adjusted prevalence ratios (aPR), a significance level of 5% and a confidence interval (CI) of 95% were considered. Statistical analysis was performed using SPSS v 21.0 (Chicago, IL, USA).

## 3. Results

The study results showed that 90 (9.4%) participants completely adhered to the social distancing measures (not leaving home), 503 (52.4%) were leaving home only for essential nonwork activities, 324 (33.7%) were leaving home also for work activities, and 43 (4.5%) did not adhere to social distancing measures. In June 2020, 38 (4%) participants were classified as not active, 112 (11.7%) were insufficiently active B, 69 (7.1%) were insufficiently active A, 371 were active (38.6%), and 371 (38.6%) were very active. In addition, 341 (35.5%) had reduced their physical activity levels, and 620 (64.5%) had not reduced their activity level. Education level data showed 10 participants (1%) had completed middle school, 49 (5.1%) had completed high school, 389 (40.5%) had completed college level, and 513 (53.4%) had completed graduate level. Family income was lower than 1 minimum wage for 17 (1.8%) participants, minimum wage between 1 and 2 for 40 (4.2%) participants, between 3 and 6 for 322 (33.5%) participants, and more than 6 minimum wages for 582 (60.6%) participants.

The level of physical activity of the participants was significantly reduced during the pandemic period (*p* < 0.001). The median and interquartile interval prior to the pandemic period was [5(4–5)], and during the pandemic period it was [4(4–5)]. All the physical activity levels presented significant difference in frequency of responses between pre- and during pandemic periods. There was a significant reduction in very active participants (X^2^(1) = 153.032; *p* < 0.001) and an increase in active (X^2^(1) = 25.390; *p* < 0.001), irregularly active A (X^2^(1) = 5.250; *p* < 0.001), irregularly active B (X^2^(1) = 50.239; *p* < 0.001), and not active (X^2^(1) = 12.971; *p* < 0.001) participants (Table 2). According to the absolute values, 555 participants maintained their physical activity levels; 341 participants decreased their physical activity level, and only 65 increased their physical activity level.

The level of association between the decrease in physical activity level during the pandemic period and age, gender, restriction level, education level, family income, daily working hours, and the physical activity level adopted prior to the pandemic period were also studied. The results are presented in Table 3.

Restriction level, age, gender, education level, and daily working hours were not associated with the physical activity reduction during the pandemic period. According to the age, people who reduced their level of physical activity were 38 (30–48) years old and people who did not reduce their level of physical activity were 38 (31–47) years old. For daily working hours, people who reduced their level of physical activity worked for 8 (6–9) hours per day, and people who did not reduce their level of physical activity worked for 8 (6–10) hours per day. On the other hand, the physical activity level adopted prior to the social distancing period and the family income presented a significant association (*p* < 0.001 and *p* = 0.001, respectively). The proportion of people who reduced their physical activity level was greater among those who were very active than among those who were active (adjusted prevalence ratio 0.65 [CI: 0.52–0.80]) or among those who were insufficiently active B (adjusted prevalence ratio 0.39 [CI: 0.18–0.82]). According to family income, the proportion of people who reduced their physical activity level was greater among those who received less than 1 minimum wage than those who received between 3 and 6 minimum wages (adjusted prevalence ratio 0.50 [CI: 0.35–0.70]), or among those who received more than 6 minimum wages (adjusted prevalence ratio 0.56 [CI: 0.40–0.79]).

## 4. Discussion

The main findings from the present study were (1) physical activity levels decreased significantly during the pandemic period; (2) the proportion of people who reduced their physical activity level was greater among those who were very active than those who were active or among those who were classified as insufficiently active B (3) the proportion of people who reduced their physical activity level was greater among those whose family received less than 1 minimum wage than those whose family received more than 3 minimum wages; and (4) gender, age, restricted rules of social distancing, education level, and daily working hours did not affect physical activity levels during the pandemic period.

During the pandemic period the physical activity level was significantly reduced. The reduction in physical activity took place for those who were classified initially as very active (60.2%). There was a significant increase in the number of active, irregularly active A, irregularly active B, and inactive participants during the pandemic period. Although some participants increased the physical activity level (n = 65), a large majority reduced their level of physical activity (n = 341) or did not change it(n = 555). Therefore, the increase in the number of active people does not mean that the people have become more active, but the large reduction in the number of very active people (n = 208) has resulted in an increase in the number of people in all other classifications of lower levels of physical activity.

Pecanha et al., (2020) reported on how social distancing resulted in a profound decrease in moderate-to-vigorous physical activity levels. Despite previous research discussing the negative impact of social distancing on physical activity level [3,28], a few studies measured the impact of the COVID-19 pandemic on physical activity behavior [23]. Lesser and Nienhuis (2020) developed a study with this aim in a Canadian population [29]. The authors also found a large percentage of participants who became less active during the pandemic period. However, contrary to our findings, they reported a greater proportion of people who were inactive prior to the pandemic period reporting less physical activity since the COVID-19 outbreak. This situation is quite worrying because a higher level of physical inactivity has been seen to be consistently associated with a higher incidence of chronic and mental disease [6].

A significant association was found between the change in physical activity and family income; a higher proportion of people who decreased their physical activity level were among those whose family received less than 1 minimal wage. These findings were concluded as lower physical activity levels was expected for those with a lower family income [30]. Silva et al., (2018) investigated the reasons why people considered being inactive, and 10% of them referred to financial problems as the most important reason. It is possible to imagine that the rate of physical inactivity among the poorest worsens even more, since the COVID-19 pandemic has also been accompanied by an economic crisis and an increase in the unemployment rate [31]. In Brazil, in addition to the economic crisis, there is also a political crisis that may also be contributing to the reduction of the physical activity level. Brazil’s president, Jair Bolsonaro, has discouraged physical distancing measures along with the use of face masks since the beginning of the pandemic, which is contrary to the recommendations of health organizations [32]. In the same context, the Brazilian government reported that it would change its method of sharing information about the pandemic. The presentations of accumulated cases and deaths were stopped on 6 June 2020, and the Supreme Court of Brazil determined that the federal government should present these data [33]. However, doubt about the transparency of the data remained. This has led to an increased sense of insecurity among the people regarding COVID-19 [34] and the guidelines provided by health organizations.

No significant association was found between change in physical activity and gender. Although the pandemic affected the level of physical activity of both men and women equally, around the world, women were less active than the men, as approximately 28% of male and 34% of female adults were insufficiently active in recent years [6]. This high percentage of insufficiently active women was already a concern before the pandemic [35], and now, with the decrease in physical activity levels, the concern with women’s physical inactivity was even greater. Physical inactivity is linked with the premature development of several chronic diseases such as metabolic syndrome, hypertension, or diabetes [36,37]. This is a worrying situation for the female population since metabolic syndrome-related cardiovascular disease is more prevalent in women than in men [38].

The results showed no association between the change in physical activity level and age. Although young people are at a lower risk for noncommunicable diseases than older adults, reports point to a high incidence of symptoms of anxiety and depression in young people during the COVID-19 pandemic [39], a condition that can be aggravated by physical inactivity [40]. Therefore, physical activity should be encouraged as a way to promote health in all age groups.

This body of knowledge can inform evidence-based policies to be implemented to counteract the impact of the reduction in physical activity level and sedentary behavior during the COVID-19 outbreak, thereby alleviating the global burden of noncommunicable diseases [28]. Although there are some previous published studies that recommended exercise that can be performed at home [19,41,42], in Brazil there are no public strategies at this time.

This study had some limitations. First, the respondents needed to remember what their habits were like prior to the COVID-19 pandemic. Secondly, they also needed to access to questionnaire on the internet. Therefore, a higher socioeconomic status was naturally selected among the studied sample. A large percentage of respondents were very active prior to the pandemic, and this characteristic may not be representative of the general population. There is a possibility that other factors are associated with reduced physical activity, such as racism, discrimination, or sexism that were not evaluated in this study. Therefore, the authors suggest caution in extrapolating from the findings.

## 5. Conclusions

Despite the recommendation to maintain physical activity at home during the COVID-19 pandemic, the physical activity level of Brazilians who lived in São Paulo significantly decreased, mainly among those who were more active and those families who received less than 1 minimal wage. For this reason, home-based physical activity programs should be strongly recommended so people could continue remaining active even while respecting the rules of social distancing.

## Figures and Tables

**Table 1 ijerph-18-11397-t001:** Descriptive characteristics of participants.

Variables	Men N = 380	Women N = 581	*p*-Value
Age (years)	40 (32–48)	37 (30–47) *	*p* < 0.013 *
Body mass (kg)	79.5 (72–88)	63 (56–72) *	*p* < 0.001 *
Height (cm)	177 (171–181)	163 (160–168) *	*p* < 0.001 *
BMI (kg/m^2^)	25.1 (23.5–27.7)	23.5 (21.4–26.3)	*p* < 0.001 *

Values were expressed as median (interquartile interval). BMI, body mass index. * Men were significantly different from women.

**Table 2 ijerph-18-11397-t002:** Physical activity level adopted prior and during the pandemic period.

	Prior Pandemic Period		During Pandemic Period		
	**Number of Volunteers**	**Percentage of Volunteers (%)**	**Number of Volunteers**	**Percentage of Volunteers (%)**	***p*-Values ***
Not active	16	1.7	38	4.0	<0.001
Irregularly active B	37	3.9	112	11.7	<0.001
Irregularly active A	47	4.9	69	7.2	0.021
Active	282	29.3	371	38.6	<0.001
Very active	579	60.2	371	38.6	<0.001

* Comparison of the frequency of responses in each category between the two evaluation moments.

**Table 3 ijerph-18-11397-t003:** Crude and multivariable analysis with estimates of the crude prevalence ratios (cPR) and adjusted prevalence ratios (aPR) for the factors associated with decreased physical activity level.

Variable	%	Crude Analysis		Multivariable Analysis	
		cPR [CI 95%]	*p*-Value ^#^	aPR [CI 95%]	*p*-Value ^#^
Restriction level			0.460		0.603
Did not adhere to social distancing	4.5	1		1	
Leaving home for working	33.8	0.77 [0.53–1.14]		0.77 [0.52–1.14]	
Leaving home only for essential nonwork activities	52.4	0.87 [0.60–1.27]		0.84 [0.58–1.22]	
Completely adhered to the social distancing isolation	9.4	0.90 [0.58–1.40]		0.83 [0.53–1.30]	
Age (years)		1 [1–1.01]	0.436	1 [1–1.01]	0.227
Gender			0.348		0.319
Women	60	1		1	
Men	40	0.92 [0.77–1.10]		0.91 [0.76–1.09]	
Physical activity level **			<0.001		<0.001
Very active	60.2	1		1	
Active	29.3	0.64 [0.52–0.80]		0.65 [0.52–0.80]	
Irregularly active A	4.9	0.76 [0.49–1.16]		0.75 [0.49–1.16]	
Irregularly active B	3.9	0.39 [0.18–0.80]		0.39 [0.18–0.82]	
Not active	1.7	0.00 (-) *		0.00 (-) *	
Education level			0.048		0.060
Middle level	1	1		1	
High school level	5.1	0.76 [0.32–1.82]		0.96 [0.40–2.29]	
College level	40.5	1.01 [0.47–2.19]		1.35 [0.62–2.90]	
Graduate level	53.4	0.80 [0.37–1.72]		1.08 [0.50–2.35]	
Family income			0.007		0.001
Less than 1 minimum wage	1.8	1		1	
Between 1 and 2 minimum wages	4.2	0.58 [0.34–0.98]		0.62 [0.38–1.02]	
Between 3 and 6 minimum wage	33.5	0.51 [0.35–0.75]		0.50 [0.35–0.70]	
More than 6 minimum wages	60.6	0.56 [0.38–0.80]		0.56 [0.40–0.79]	
Daily working hours (hours per day)		0.98 [0.95–1.01]	0.231	0.99 [0.96–1.02]	0.624

^#^ Significance level for the Poisson regression. * (-) inaccurate. ** Physical activity level prior to the pandemic period. Irregularly active A, those who perform physical activity, but it is insufficient to be classified as active because it does not comply with the recommendations regarding frequency or duration; irregularly active B, those who perform physical activity, but it is insufficient to be classified as irregularly active A, because it does not comply with either the frequency or duration recommendations.

## Data Availability

The data presented in this study are available on request from the corresponding author. The data are not publicly available due to privacy and ethical reasons.

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
