# Peer review of "Factors Associated with Reduction in Physical Activity during the COVID-19 Pandemic in São Paulo, Brazil: An Internet-Based Survey Conducted in June 2020"

_ijerph, 2021, doi:10.3390/ijerph182111397_

Round 1

Reviewer 1 Report

This current study is missing a hypothesis, but I deduce that the authors hypothesized that physical activity would decrease during the lockdowns associated with COVID-19?  I will provide only overall comments as the manuscript needs to be improved:

  1. Your title should reflect both that the study is only limited to Brazil and clarify whether you conducted the online survey in June 2020 or 2021?
  2. You need to clarify the geographic scope of the 1400 participants (did they come from urban centers? which ones?)
  3. You need to clarify race/ethnicity data of participants
  4. You need much more work on the confounding variables (some of which you identify) such as the built-environment/landscape of the cities/locations where the participants are located.  
  5. The participants section (2.2) has too much detail of the survey and not enough description of the participants.  Please remove the data on survey and recruitment and place that in the method section where you are discussing the instrument. It would help to have an example of your survey as an appendix.
  6. Please consider getting an English-language editor to assist you with grammar consistency and use "pandemic" (not epidemic) when discussing COVID-19.
  7. You need to include more information in the background on Brazil's handling of COVID-19 and ensure that your audience understands the political environment since the lack of strict protocols and health promotion measures may contribute to lack of physical activity.
  8. Add more background on the possible social and cultural determinants of sedentary behavior in Brazil (i.e., racism/discrimination can lead to lack of physical activity).
  9. Improve the description of your methods. If this is a small part of a larger study, that needs to be transparent earlier on in the manuscript (include it in the abstract) so that you are providing full disclosure.
  10. I think your study is an important one, but needs to be much more transparent and grounded in the Brazilian context.

Author Response

Submission ID ijerph-1410482

Title: Factors associated with reduction in physical activity during the COVID-19 pandemic in Brazil: An Internet-Based Survey conducted in June 2020 (former title: Factors associated with reduction in physical activity during the COVID-19 pandemic: An Internet-Based Survey )

05-October-2021

Paul B. Tchounwou 

Editor-in-Chief

Dear Editor,

Thank you for the opportunity to submit a revised version of this manuscript. We also thank the two reviewers for their constructive comments and insightful observations. We have made a number of extensive revisions and considerably improved the major sections of the paper. We are sending point-by-point answers to the comments. The changes in the text were made using the track change mode in MS Word. The manuscript has improved substantially, and we hope it is now suitable for publication in the International Journal of Environmental Research and Public Health.

Reviewer #1

This current study is missing a hypothesis, but I deduce that the authors hypothesized that physical activity would decrease during the lockdowns associated with COVID-19?  I will provide only overall comments as the manuscript needs to be improved:

Answer: Thank you about your comments. Hypothesis is included in the end of the introduction section (the last paragraph). “We hypothesized, firstly, that even respecting the recommendation to maintain physical activity at home, the physical activity level decreased significantly in this pandemic period. We also hypothesized, secondly, that this reduction would be greater among those adhering to more restricted rules of social distancing isolation; those who presented a more vigorous physical activity level prior to the pan-demic period; those with longer daily working hours; and those with lower education level and family income.”

  1. Your title should reflect both that the study is only limited to Brazil and clarify whether you conducted the online survey in June 2020 or 2021?

Answer: We totally agree with the expert reviewer and the title has been changed including your suggestions.  New title: Factors associated with reduction in physical activity during the COVID-19 pandemic in Brazil: An Internet-Based Survey conducted in June 2020

  1. You need to clarify the geographic scope of the 1400 participants (did they come from urban centers? which ones?)

Answer: Thank you about your suggestion. Individuals from all the 26 Brazilian states and the Federal District answered the questionnaire, and 68% of the sample was from São Paulo city.

  1. You need to clarify race/ethnicity data of participants

Answer: Unfortunately, we have no data about race/ethnicity of participants. We have decided to include this as study limitation. Please let us know if this explanation does not resolve your doubts in this matter.

  1. You need much more work on the confounding variables (some of which you identify) such as the built-environment/landscape of the cities/locations where the participants are located.  

Answer: Thank you for calling our attention to this point. The sample consisted of participants from all Brazilian states, whose environmental characteristics are quite diverse. However, 68% of the sample was from São Paulo, which is a large urban center with a population of 12.3 million inhabitants and which has large numbers of parks and sports centers.

  1. The participants section (2.2) has too much detail of the survey and not enough description of the participants.  Please remove the data on survey and recruitment and place that in the method section where you are discussing the instrument. It would help to have an example of your survey as an appendix.

Answer: Thank you about your constructive comment. Data on participants’ recruitment, the questionnaire and the participants were separated to clarify the entire materials and methods section and meet with your expectation.

  1. Please consider getting an English-language editor to assist you with grammar consistency and use "pandemic" (not epidemic) when discussing COVID-19.

Answer: Thank you for calling our attention to this point. Pandemic has been used when discussing COVID-19.

  1. You need to include more information in the background on Brazil's handling of COVID-19 and ensure that your audience understands the political environment since the lack of strict protocols and health promotion measures may contribute to lack of physical activity.

Answer: We totally agree with you. “Particularly in Brazil, in addition to the economic crisis, there is also a political crisis that may also be contributing to the reduction of the population physical activity level. Brazil’s president, Jair Bolsonaro, since the begin of the COVID-19 pandemic, discourage physical distancing measures along with the use of face masks, contrary to the recommendations of health organizations. In the same context, the Brazilian government has reported that it would change its method of sharing information about the pandemic. Accumulated cases and deaths presentation were stopped on June 6, 2020, and the Supreme Court of Brazil determined that the federal government should present these data. However, doubt about the transparency of the data remained. This has led to an increased sense of insecurity amongst the Brazilian citizens regarding the COVID-19 disease [34] and regarding the guidelines provided by health organizations.” This sentence has been included in the discussion section in order to clarify and meet with your expectation.

  1. Add more background on the possible social and cultural determinants of sedentary behavior in Brazil (i.e., racism/discrimination can lead to lack of physical activity).

Answer: Thank you about your suggestion. Other possible factors associated with the physical activity reduction during the pandemic period, such as racism, discrimination, or sexism, and other not evaluated in the present study, have been included as a study limitation. Please let us know if this explanation does not resolve your doubts in this matter.

  1. Improve the description of your methods. If this is a small part of a larger study, that needs to be transparent earlier on in the manuscript (include it in the abstract) so that you are providing full disclosure.

Answer: Thank you for calling our attention to this point. The description of our methods has been further detailed in the methods section so that the reader can fully understand what was done. A more detailed information also was included in the abstract.

  1. I think your study is an important one, but needs to be much more transparent and grounded in the Brazilian context.

Answer: Several changes have been done in order to improve the aspects that you have pointed. Please let us know if the changes made have met your expectations.

Reviewer 2 Report

This manuscript presents factors associated with reduction of physical activity in first part of 2020 due to the Covid-19 pandemic. Methods are based on digital tools. 

I have only one major objection after reading the manuscript: In the discussion section (row257->270) it is found that the most active people have the greatest reduction in physical activity. This is contrary to the findings of other studies. In my opinion, I try to visualise the categories of activity as a series of "water barrels" with different heights but connected. If the highest exercise category (highest water barrel) looses content, the people (water) are transfered to the next barrel (slightly lower) and so on...  The lower categories are always filled with people from higher levels and I would like to see more of a discussion on how to follow a specific group of people (paired tests). There is a lot of statistic methods used but I believe this outcome is lost in the discussion part. 

Some other suggestions:

  • Try to fit Table 1 into one page for better overview.
  • State the year 2020 also on row 111 and 116.
  • On row 170 "increase" is mentioned - with paired data one should be able to see if subgroups had an increased level of exercise. A hypotheses could be that when there is nothing else to do in society I spend my time exercising at home.... 

Author Response

Submission ID ijerph-1410482

Title: Factors associated with reduction in physical activity during the COVID-19 pandemic in Brazil: An Internet-Based Survey conducted in June 2020 (former title: Factors associated with reduction in physical activity during the COVID-19 pandemic: An Internet-Based Survey )

05-October-2021

Paul B. Tchounwou 

Editor-in-Chief

Dear Editor,

Thank you for the opportunity to submit a revised version of this manuscript. We also thank the two reviewers for their constructive comments and insightful observations. We have made a number of extensive revisions and considerably improved the major sections of the paper. We are sending point-by-point answers to the comments. The changes in the text were made using the track change mode in MS Word. The manuscript has improved substantially, and we hope it is now suitable for publication in the International Journal of Environmental Research and Public Health.

Reviewer #2

This manuscript presents factors associated with reduction of physical activity in first part of 2020 due to the Covid-19 pandemic. Methods are based on digital tools. 

I have only one major objection after reading the manuscript: In the discussion section (row257->270) it is found that the most active people have the greatest reduction in physical activity. This is contrary to the findings of other studies. In my opinion, I try to visualise the categories of activity as a series of "water barrels" with different heights but connected. If the highest exercise category (highest water barrel) looses content, the people (water) are transfered to the next barrel (slightly lower) and so on...  The lower categories are always filled with people from higher levels and I would like to see more of a discussion on how to follow a specific group of people (paired tests). There is a lot of statistic methods used but I believe this outcome is lost in the discussion part. 

Answer: Thank you about your constructive comment. Table 2 shows a comparison between the number of participants who presented each physical activity level prior and during the pandemic period. All the physical activity levels presented a significant difference between the two periods, however only the very active group presented a significant reduction (36%). On the other hand, there were a significant increase in the number of active, irregularly active A, irregularly active B and inactive participants during the pandemic period. Despite some participants increased the physical activity level (n=87), a large majority decrease (n=506) or did not change (n=807). Therefore, the increase in the number of active people does not mean that the people have become more active, but rather that, the large reduction in very active people number (n=310) has resulted in an increase in the number of people in all other classifications of lower levels of physical activity. Please let us know if this explanation does not solve your doubt in this matter.

  • Try to fit Table 1 into one page for better overview.

Answer: Sorry, but this is a very large Table. Probably in the final pdf version this problem can be solved.  

  • State the year 2020 also on row 111 and 116.

Answer: Thank you for calling our attention to this point. The year has been included.

  • On row 170 "increase" is mentioned - with paired data one should be able to see if subgroups had an increased level of exercise. A hypotheses could be that when there is nothing else to do in society I spend my time exercising at home.... 

Answer: Yes, there are some participants who increase the physical activity level, however this number was very low (n=87) compared with those who decrease (n=506) the physical activity level. This data was included in the discussion section.

Round 2

Reviewer 1 Report

Thank you for the revisions, and responses to the comments. The added information is critical for a reviewer to evaluate your hypothesis, methods and results.  You have an important study here, but need much more rigor in your methods and data analysis.  For example, you cannot draw national conclusions if your sample is predominately from one city in Brazil. I recommend that since 68% of your sample is from Sao Paolo, you limit your analysis to that city and eliminate the other data from your analysis/results/conclusions.  You cannot make any definitive claims about an entire country when such a large portion of your sample is from only 1 city.  For this manuscript, you should focus on that dataset only and report on impacts of PA from Sao Paolo.  I recommend revising accordingly for resubmission.  As noted earlier, your manuscript would benefit from an English editor to resolve grammar errors. I send you energy to make the revisions as you have a potentially great paper which needs fine-tuning.

Author Response

Submission ID ijerph-1410482

Title: Factors associated with reduction in physical activity during the COVID-19 pandemic in São Paulo - Brazil: An Internet-Based Survey conducted in June 2020 (former title: Factors associated with reduction in physical activity during the COVID-19 pandemic in Brazil: An Internet-Based Survey conducted in June 2020 )

23-October-2021

Paul B. Tchounwou 

Editor-in-Chief

Dear Editor,

Thank you for the opportunity to resubmit a revised version of this manuscript. We also thank the reviewer for their comment. Analysis of the data was rebuilt only with participants from Sao Paulo. In addition, an English review also was performed.  We are sending point-by-point answers to the comments. The changes in the text were made using the track change mode in MS Word. The manuscript has improved substantially, and we hope it is now suitable for publication in the International Journal of Environmental Research and Public Health.

Reviewer #1

Thank you for the revisions, and responses to the comments. The added information is critical for a reviewer to evaluate your hypothesis, methods and results.  You have an important study here, but need much more rigor in your methods and data analysis.  For example, you cannot draw national conclusions if your sample is predominately from one city in Brazil. I recommend that since 68% of your sample is from Sao Paolo, you limit your analysis to that city and eliminate the other data from your analysis/results/conclusions.  You cannot make any definitive claims about an entire country when such a large portion of your sample is from only 1 city.  For this manuscript, you should focus on that dataset only and report on impacts of PA from Sao Paolo.  I recommend revising accordingly for resubmission.  As noted earlier, your manuscript would benefit from an English editor to resolve grammar errors. I send you energy to make the revisions as you have a potentially great paper which needs fine-tuning.

Answer: Thank you about your constructive comment. Participants who were not from São Paulo were excluded from the study as suggested by you. This information was included in the text. The sample consisted of 961 people. All data analysis was rebuilt, and all results were updated in the text. It is noteworthy that the variables that were associated with the change in physical activity level did not change (prepandemic physical activity level and family income). Thus, the conclusions of the study remain the same, but it was emphasized that the results refer to the population of São Paulo. The manuscript was revised by an English native speaker in order to correct typos and grammar mistakes. Please let us know if the changes do not met your expectations.